# Preclinical Efficacy of a Trivalent Human FcγRI-Targeted Adjuvant-Free Subunit Mucosal Vaccine against Pulmonary Pneumococcal Infection

**DOI:** 10.3390/vaccines8020193

**Published:** 2020-04-23

**Authors:** Sudeep Kumar, Raju Sunagar, Edmund J. Gosselin

**Affiliations:** 1Department of Immunology and Microbial Diseases, Albany Medical College, Albany, NY 12208, USA; kumars@amc.edu; 2Ella Foundation, Genome Valley, Hyderabad 500078, India; rajus@ellafoundation.org

**Keywords:** mucosal vaccine, subunit vaccine, pneumococcal vaccine, antigen-presenting cell targeting, mucosal adjuvant, intranasal vaccine, Fc gamma receptor

## Abstract

Lack of safe and effective mucosal adjuvants has severely hampered the development of mucosal subunit vaccines. In this regard, we have previously shown that immunogenicity of vaccine antigens can be improved by targeting the antigens to the antigen-presenting cells. Specifically, groups of mice immunized intranasally with a fusion protein (Bivalent-FP) containing a fragment of pneumococcal-surface-protein-A (PspA) as antigen and a single-chain bivalent antibody raised against the anti-human Fc-gamma-receptor-I (hFcγRI) elicited protective immunity to pulmonary *Streptococcus pneumoniae* infection. In order to further enhance the immunogenicity, an additional hFcγRI-binding moiety of the single chain antibody was incorporated. The modified vaccine (Trivalent-FP) induced significantly improved protection against lethal pulmonary *S. pneumoniae* challenge compared to Bivalent-FP. In addition, the modified vaccine exhibited over 85% protection with only two immunizations. Trivalent-FP also induced *S. pneumoniae-*specific systemic and mucosal antibodies. Moreover, Trivalent-FP also induced IL-17- and IL-22-producing CD4^+^ T cells. Furthermore, it was found that the hFcγRI facilitated uptake and presentation of Trivalent-FP. In addition, Trivalent-FP also induced IL-1α, MIP-1α, and TNF-α; modulated recruitment of dendritic cells and macrophages; and induced CD80/86 and MHC-II expression on antigen presenting cells.

## 1. Introduction

Mucosal surfaces are prone to infection by many pathogens, and adequate protection of mucosal surfaces require the presence of effective local immune response [1,2,3,4,5,6]. Previous reports suggest that immunization by mucosal route (such as intranasal or oral immunization) can achieve superior mucosal immune response compared to the parenteral route (such as intradermal or intramuscular immunization) [7]. Although the mucosal vaccines such as typhoid, cholera, and flu vaccines are effective at eliciting mucosal immunity, these are based on live-attenuated or inactivated whole cell vaccine platforms. However, due to potential reversion to pathogenic forms or incomplete inactivation, the live attenuated and whole cell-inactivated vaccines pose significant safety concerns [8]. On the other hand, subunit vaccines offer a much safer alternative to the whole cell-based vaccines as they rely on purified and/or detoxified microbial components [8]. However, the increased safety profile of subunit vaccines comes at a cost of reduced efficacy. Thus, subunit vaccines require adjuvants and often times also require multiple immunizations to evoke an adequate immune response [8]. Importantly, the propensity of tolerogenic immunity at mucosal sites and lack of safe and effective mucosal adjuvants makes mucosal immunization more challenging [9].

With over 90 known serotypes, *Streptococcus pneumoniae* remains the leading cause of community acquired pneumoniae. In addition, younger and immunocompromised populations are more vulnerable to infection. Furthermore, *S. pneumoniae* causes over 1 million deaths per year in children aged less than 5 years [10]. To date, two polysaccharide-based subunit vaccines are available to combat select serotypes. However, use of these 13 and 23 serotype vaccines cause serotype replacement in the vaccinated population [11]. This results in a surge of non-vaccine serotypes within the vaccinated population. Thus, new approaches to pneumococcal vaccines are required, which can generate protection over multiple serotypes. The pneumococcal surface protein A (PspA) has been known to induce cross-serotype protection against *S. pneumoniae* [12]. Thus, not only is *S. pneumoniae* a significant global health problem, it also expresses a well-defined protective protein antigen, making it a particularly suitable model to test and optimize our mucosal subunit vaccine platform. 

Previously we have shown that targeting vaccine antigens to antigen-presenting cells (APCs) eliminates the need for adjuvants. By genetically fusing a bivalent single-chain variable fragment-based antibody (Bivalent anti-human-Fc-gamma-receptor-I (FcγRI)), which specifically recognizes human-FcγRI, to a pneumococcal antigen (PspA), a fusion protein named Bivalent-FP was obtained (Figure 1A). Bivalent-FP induces systemic and mucosal antibodies and protection against pulmonary *S. pneumoniae* infection by intranasal immunization. However, this vaccine requires at least three immunizations to achieve an adequate level of protection [13]. To further improve the efficacy of this human-FcγRI-targeted vaccine, we added an additional human-FcγRI-binding moiety to Bivalent-FP. The modified vaccine is called trivalent anti-human-FcγRI-PspA (Trivalent-FP) (Figure 1B).

In this investigation, we first compared the efficacy of our novel Trivalent-FP to our earlier vaccine, Bivalent-FP. After demonstrating that Trivalent-FP was superior at inducing protection against *S. pneumoniae* versus Bivalent-FP, we focused this investigation on evaluation of the capacity of Trivalent-FP to induce mucosal immune response. Specifically, we evaluated the secretory antibody response, which plays an important role in restricting bacterial invasion through the mucosa. Apart from the secretory antibodies, cytokines produced by T helper-17 (Th17) and T helper-22 (Th22) cells have been shown to play important roles in protection against several *S. pneumoniae* strains [14,15,16,17]. Specifically, IL-17 and IL-22 produced by these cells together induce secretion of chemokines and antimicrobial peptides, as well as recruitment of neutrophils, which promote bacterial clearance [18]. Furthermore, IL-22 plays important roles in restoring epithelial barrier function by inducing epithelial cell division [19]. Therefore, we investigated the Th17 and Th22 responses elicited by Trivalent-FP. Moreover, because it was evident in our previous study [13] and confirmed in this study that neither Bivalent-FP nor Trivalent-FP requires traditional adjuvant for the induction of a protective immune response, we also investigated whether Trivalent-FP can induce adjuvant-like effects. 

## 2. Materials and Methods

### 2.1. Mice 

C57BL/6 (WT) mice were obtained from Taconic Laboratories (Germantown, NY, USA). The transgenic mice designated as Tg in the manuscript express human Fc-gamma-receptor-I [20]. This strain was a generous gift from Medarex Inc. (Bloomsbury, NJ). The heterozygous Tg mice were maintained by breeding with wild type (WT) C57BL/6. A PCR-based genotyping method was used to distinguish the heterozygous Tg mice from the WT littermates. The WT littermates were utilized as the WT controls. All mice were housed in the animal resources facility of Albany Medical College under pathogen-free conditions. Mice were given food and water ad libitum during the entirety of the experiment. 

### 2.2. Ethics Statement

All experiments that used mice were conducted according to the standards of Institutional Animal Care and Use Committee (IACUC) of Albany Medical College, Albany, NY, USA. The institutional standard follows the National Institute of Health, USA, guidelines. Prior to the start of the study, the specific animal protocols were approved by the IACUC of Albany Medical College.

### 2.3. Vaccine Preparation

Both Bivalent-FP and Trivalent-FP were manufactured by GenScript (Piscataway, NJ, USA). Their maps are described in Figure 1A,B. The genetic code of both vaccines were optimized for expression in Chinese hamster ovary (CHO) cells and the relevant DNA fragments were synthesized by artificial gene synthesis. The complete gene sequence was then inserted into the mammalian expression plasmid pTT5 between EcoRI and HindIII restriction sites. Proteins were obtained from supernatants of transfected CHO cells and were purified by two step process by fast-performance-liquid-chromatography and using HisTrap FF Crude column (GE Life Sciences, Pittsburgh, PA, USA) in step one and HiLoad 26/60 Superdex 200 column (GE Life Sciences) in step two. The purified protein was stored in phosphate buffered saline (PBS) (pH 7.2) at −80°C until use. 

### 2.4. Immunization and Challenge

Groups of mice were immunized on days 0 and 21 via the intranasal route. Briefly, mice were anesthetized by intra-peritoneal (i.p.) injection of 20% ketamine plus 5% xylazine, and subsequently 20 µL PBS (vehicle) or 208 pmol Bivalent- or Trivalent-FP (dissolved in 20 µL of PBS) were administered via intranasal route. Immunized mice were challenged at 2 weeks post-booster immunization with 2 × 10^6^ colony forming units (CFUs) of *S. pneumoniae* (serotype type 3, strain A66.1) suspended in 40 µL PBS, via the intranasal route. Following infection, survival of animals was recorded once daily for 21 days. 

### 2.5. Serum and Bronchoalveolar Lavage Collection

For serum isolation, mice were bled via submandibular puncture using a 4 mm lancet, and 100 to 200 µL blood was collected from each mouse. The blood was incubated at 37 °C for 1 h and centrifuged at room temperature (RT) for 5 min at 8000 rpm. The clear supernatant was then collected in fresh tubes and stored at −20 °C until further analysis. 

For collection of bronchoalveolar lavage (BAL), mice were euthanized, and the lungs were flushed 3× with 1mL PBS with tracheal insertion of an 18-gauge needle. The lung washes were centrifuged at 5000 rpm for 5 min, and the clear supernatant (BAL) was stored at −20 °C until further analysis.

### 2.6. Quantification of Bacterial Burden

Following immunization with 20 µL PBS (vehicle) or 208 pmol Bivalent-FP or Trivalent-FP (dissolved in 20 µL of PBS), mice were challenged with 0.5 × 10^6^ CFU of *S. pneumoniae* (serotype type 3, strain A66.1) suspended in 40 µL PBS, via the intranasal route. Mice were euthanized at various intervals and lungs were collected aseptically in PBS. Lungs were homogenized with a Mini-Bead-Beater-8 (BioSpec Products, Bartlesville, OK, USA) using 1 mm zirconia/silica beads. The homogenates were then serially diluted (10-fold in each step) up to 1:10^7^ in sterile PBS, and 10 µL of each dilution was spotted onto Trypticase Soy Agar plates with 5% sheep blood (BD Biosciences, Franklin Lakes, NJ, USA) in duplicate and incubated at 37 °C for 24 h. Number of colonies (CFUs) on the plates were counted and expressed as CFU per milliliter for respective samples. Similarly, blood was also isolated by submandibular vein puncture and collected in tubes containing 50 µL 4% sodium citrate as anticoagulant. Blood samples were similarly diluted and plated for enumeration of *S. pneumoniae* CFUs.

### 2.7. Antibody Titer Determination

*S. pneumoniae-*specific Ab levels in sera and BAL were measured by ELISA. Briefly, ELISA plates (Corning, Corning, NY, USA) were coated with 50 µL of *S. pneumoniae* (5 × 10^7^ CFU/mL) in carbonate buffer (4.3 g/L sodium bicarbonate (Sigma-Aldrich, St Louis, MO, USA) and 5.3 g/L sodium carbonate (Sigma-Aldrich) (pH 9.4) for 16 h at 4 °C. The plates were then washed with washing buffer (PBS (Sigma-Aldrich) containing 0.1% Tween 20 (Sigma-Aldrich)) and incubated at 4 °C for 1 h with 200 µL of 5% bovine serum albumin (BSA) (in PBS). Serial threefold dilutions of sera/BAL (starting with 1/15) were added to the plates (50 µL/well) and incubated for 2 h at 4 °C. Following the completion of incubation, plates were washed 3× with washing buffer. Subsequently, alkaline-phosphatase conjugated anti-mouse Abs specific for IgG, IgA, IgG1, and IgG2c (Sigma-Aldrich) was added and incubated for 1 h at 4 °C. The plates were then washed 3× with washing buffer. Subsequently, 100 µL of alkaline phosphatase substrate (Sigma-Aldrich) was added and incubated for 2 h at room temperature (RT) and the absorbance at 405 nm was recorded using a microplate reader (Molecular Devices, Sunnyvale, CA, USA). Ab titers were deduced on the basis of the four-parameter nonlinear regression model of (log) titers vs. absorbance, wherein the IC_50_ value was considered as titer.

### 2.8. Lung Cell Isolation and CD4^+^ T Cell Analysis

Six weeks following immunization, mice were infected with 0.5 × 10^6^ CFUs of *S. pneumoniae* in 40 µL PBS via the intranasal (i.n.) route. At 6 h and 24 h post-infection, mice were euthanized, and their lungs were isolated and kept in serum-free RPMI (1 mL/lung) until all the lungs were harvested. Lungs were then minced into very small pieces (< 2mm) using scissors. The minced lung fragments were then re-suspended in digestion buffer (1 mL/lung; Liberase 250 µg/mL (Sigma-Aldrich), DNAseI 250 µg/mL (Sigma-Aldrich), in serum free RPMI)). The lung suspension was incubated for 1 h at 37 °C with vigorous shaking. The digested lungs were then passed through 70 µm nylon mesh strainers fitted on 50 mL tubes. The lung pieces on the mesh strainer were mashed with the rubber end of a 3 mL syringe plunger and the strainer was rinsed intermittently with 2–4 mL RPMI to let the cells pass through the mesh. The filtrate was centrifuged at 1500 rpm in a swing bucket rotor for 5 min. The pellet was re-suspended in 10 mL RBC lysis buffer (ammonium chloride: 8.02 g/L, sodium bicarbonate: 0.84 g/L, and Ethylene-diamine-tetra-acetic-acid -(Na)_2_: 36 mg/L) and incubated at RT for 1 min. A total of 20 mL RPMI was then quickly added and the cells were centrifuged at 1500 rpm for 5 min. After one more washing with 20 mL RPMI, cells were stained with CD45-APC-fire-750, CD3-V450, and CD4-FITC antibodies. Following the surface staining, cells were fixed and permeabilized with cytofix/cytoperm buffer (BD Biosciences, Franklin Lakes, NJ, USA). Cells were then stained with IL-17-PE and IL-22-APC Abs and analyzed by flow cytometer.

### 2.9. Lung Cell Treatment with FP for Cytokine Release

Lung cells from WT and Tg mice were isolated as above, and 2 × 10^6^ cells in 100 µL of RPMI-1640 + 10% fetal bovine serum were treated with 50 µL PBS, 50 µg FP, or 100 ng Lipopolysaccharide. After 48 h of treatment, the supernatant was evaluated for the presence of multiple cytokines using a multiplex Bio-Plex assay. 

### 2.10. Modulation of CD64 on APCs 

Splenocytes from Tg or WT mice were treated with various amounts of FPs at 4 °C and 37 °C for 2 h, and subsequently the level of surface bound FcγRI was evaluated. Briefly, 1 × 10^6^ splenocytes were suspended in 100 µL staining buffer (1% BSA/PBS) and treated with 2 to 10 pmol/mL of FPs. Splenocytes were incubated at 4 and 37 °C for 2 h, and washed 2× with 2 mL staining buffer. Subsequently, splenocytes were stained with anti-mouse CD19-PE, anti-mouse CD11b-APC, anti-mouse CD11c-BV421, and anti-human or anti-mouse CD64-FITC and analyzed by flow cytometer (LSRII) (BD, San Jose, CA, USA). These markers were chosen to discriminate macrophages and dendritic cells (DCs). The splenocyte macrophage and DCs express CD11b and CD11c. However, B cells may also express CD11b. Thus, to exclude B cells, CD19^−^CD11b^+^CD11c^+^ gating was used. CD64^+^ Mean Fluorescence Intensity (MFI) were evaluated on the CD19^−^CD11b^+^CD11c^+^ population. Percentage of human or mouse FcγRI (CD64) MFI relative to untreated cells were calculated as: percentage CD64 MFI = (CD64 MFI of test sample/CD64 MFI of untreated sample) × 100. 

### 2.11. Antigen Presentation Assays

The PspA-specific T cell hybridoma (B6D2) [21] (1 × 10^5^ cells/well), which secretes IL-2 in response to Ag presentation, was co-cultured with macrophage (2 × 10^5^ cells/well) from either WT or Tg mice with 0.2 to 1.0 nM Trivalent-FP. The co-culture was incubated for 48 h at 37 °C in 5% CO_2_. Subsequently, supernatants were collected, and the IL-2 content was measured using the Bio-plex assay, as described below.

### 2.12. Multiplex Cytokine Analysis

Cytokines were measured using Bio-Plex assay kits (Bio-Rad, Hercules, CA, USA) following the manufacturer’s instructions. Briefly, 50 μL of a 1× mixture of magnetic beads coupled with capture Abs for capturing desired cytokines were added to separate wells of flat-bottom 96-well micro-titer plate. Beads were washed 2× with Bio-Plex wash buffer. Unless stated otherwise, all incubations were performed at RT with plates sealed with a plastic sealer and covered with aluminum foil with gentle shaking. All washing was performed with Bio-Plex washing buffer. A total of 50 μL of diluted standards, blank, or samples were added to individual wells. Plates were incubated for 30 min and washed 3× with Bio-Plex washing buffer. Then, 25 μL of 1× mixture of biotinylated detection Abs were added to each well and incubated for 30 min. Plates were washed again and 50 μL of phycoerythrin-coated streptavidin was added to each well and incubated for 10 min. Plates were then rewashed and 125 μL of assay buffer was added to each well. Plates were read via Luminex reader (Bio-Rad) and the quantity of each cytokine was deduced using a standard curve included in the assay.

### 2.13. Nasal-Associated Lymphoid Tissue Evaluation

To evaluate cell recruitment and APC activation in the nasal-associated lymphoid tissue (NALT), mice were immunized with 20 μL PBS or 20 μg Trivalent-FP (in 20 μL PBS), via the intranasal route. Then, 2 h or 24-h post-immunization, mice were euthanized and their upper palate that separates the buccal cavity and nasal passage was isolated. NALT is located bilaterally at the posterior side of the palate [22]. Each palate was immersed in 200 μL PBS and vortexed vigorously for 1 min, which allowed loosely attached cells in the NALT to be released in surrounding media (PBS). Cells in PBS were then transferred into a fresh tube, leaving behind the upper palate. Each palate was further washed with 1mL PBS 2× with 10 sec vortex each time, and the cells released in PBS were combined with the cells obtained in the previous wash step. NALT cells were centrifuged at 1500 rpm for 5 min. 

To evaluate cell recruitment in response to immunization, NALT cells were isolated at 2 h post-immunization as above, and total cells were enumerated by microscopy. Subsequently, cells were stained with CD45-APC-Fire-750, CD11b-FITC, CD11c-APC, Gr1-Pacific blue, and F4/80- PE. CD45-positive cells were further evaluated, along with DC (CD11c^+^), macrophages (CD11b^+^, F4/80^+^), and polymorphonuclear leukocytes (PMN) (Gr1^+^, SSC^high^).

To evaluate APC activation, the NALT cells were re-suspended in staining buffer and stained with CD11b-PE-Cy7, CD11c-APC, MHC-II-FITC, CD19-PE, CD80-PerCP-Cy-5.5, CD86-PerCP-Cy-5.5, and Fixable Viability Dye eFluor 780. Cells were gated on live-cells, CD19^−^, and MHC-II^+^, and MFI of MHC-II and CD80/86 was calculated.

### 2.14. Statistics 

The log-rank (Mantel–Cox) test was used for survival curves. Mann–Whitney nonparametric test was used for comparisons of different groups in the rest of the figures. All statistical analyses were performed using Graph-Pad Prizm 5 software (Graph-Pad Software, San Diego, CA, USA).

## 3. Results

### 3.1. Trivalent-FP Induced Better Protection against S. pneumoniae Challenge Compared to Bivalent-FP

To evaluate the efficacy of Bivalent-FP and Trivalent-FP, we utilized the human-FcγRI expressing transgenic mice [20] because the APC-targeting component in these vaccines recognizes the human-FcγRI and not the mouse-FcγRI. The wild type littermates generated by crossbreeding the human-FcγRI heterozygous males with C57BL/6 females, which are referred to here as WT, were utilized as controls. To begin with, groups of WT and Tg mice were immunized via the intranasal route with Bivalent-FP or Trivalent-FP. Two weeks following the prime and booster immunizations, mice were infected with a lethal dose of *S. pneumoniae*. The Tg groups receiving Trivalent-FP exhibited significantly higher protection compared to that of Bivalent-FP or PBS recipients (Figure 1C). Moreover, the protection was dependent on the presence of the human-FcγRI, as Trivalent-FP protected 85% of the Tg mice but only 14% of the WT mice. Furthermore, the involvement of human-FcγRI targeting was also evident with immunization with Bivalent-FP, as the Tg mice showed 50% protection but the WT mice showed only 14% protection (Figure 1C).

Following the survival studies, the pathogen clearance capacity of Trivalent-FP and Bivalent-FP were evaluated. A significant reduction in the colony forming units (CFUs) of *S. pneumoniae* were observed in the lungs and blood of the Tg mice immunized with Trivalent-FP. Although Bivalent-FP also caused significant reduction in the *S. pneumoniae* CFUs, Trivalent-FP induced even better bacterial clearance than Bivalent-FP. Moreover, the bacterial clearance by both Bivalent-FP and Trivalent-FP was dependent on human-FcγRI as the Tg mice receiving the FP exhibited significantly lower *S. pneumoniae* CFUs compared to the WT mice receiving the same vaccines (Figure 1D). 

### 3.2. Trivalent-FP Induced Systemic Antibody Response 

Pathogen-specific antibodies are key to resistance against *S. pneumoniae*. Thus, we evaluated the *S. pneumoniae*-specific antibody responses generated by Trivalent-FP. Eleven days following the booster immunization, sera was isolated and *S. pneumoniae*-specific IgG, IgA, IgG1, and IgG2c were evaluated. Trivalent-FP induced significantly higher levels of IgG (Figure 2A), IgA (Figure 2B), and IgG1 (Figure 2C) in the Tg group compared to the PBS- or Bivalent-FP-immunized Tg groups. Trivalent-FP also induced significantly higher levels of IgG2c compared to the PBS-immunized Tg group (Figure 2D). Moreover, the antibody response by Trivalent-FP was dependent on the presence of human-FcγRI as the Tg groups immunized with Trivalent-FP exhibited significantly higher IgG (Figure 2A), IgA (Figure 2B), and IgG1 (Figure 2C) compared to the similarly immunized WT groups. 

### 3.3. Trivalent-FP Induced Mucosal Immune Response

Given that our Trivalent-FP was more efficacious than Bivalent-FP, we reasoned that it would yield a greater overall signal relevant to cell-mediated and innate immune responses than that generated by Bivalent FP. Therefore, we focused the remainder of our studies on Trivalent-FP. 

Inducing vaccine-specific mucosal immunity is a key objective of developing this vaccine platform. In this regard, we evaluated the secretary antibodies (IgA and IgG) in the lung mucosa (that is, in the bronchoalveolar lavage) that play a key role in preventing microbial invasion through mucosal surfaces. Mice immunized with PBS or Trivalent-FP were euthanized, the BAL was isolated, and *S. pneumoniae*-specific IgA and IgG titer were evaluated. Trivalent-FP induced a significantly higher *S. pneumoniae-*specific IgA (Figure 3A) and IgG (Figure 3B) in the Tg group compared to the PBS-immunized Tg group. The mucosal antibody response by Trivalent-FP was dependent on the human-FcγRI, as the Tg Trivalent-FP immunized group exhibited significantly higher IgG and IgA compared to the respective WT groups (Figure 3A,B).

In unimmunized mice, *S. pneumoniae* causes an acute infection wherein mice become moribund within 3–4 days, and thus it is anticipated that the protective response in vaccinated mice must be elicited very early. On the other hand, we also observed that *S. pneumoniae* infection causes recruitment of Th17 and Th22 cells with delayed kinetics. In contrast to *S. pneumoniae* infection in unimmunized mice, we anticipated that the immunized mice would recruit Th17 and Th22 cells with faster kinetics. Thus, we evaluated IL-17- and IL-22-producing CD4^+^ T cells at 6 h (for immediate early) and (24 h for early response) post-infection in the lungs of mice immunized with PBS versus Trivalent-FP. 

Trivalent-FP induced IL-17 alone, IL-22 alone, as well as IL-17- and IL-22-producing CD4^+^ T cells (Figure 4A–F). Trivalent-FP induced an early (6 h) increase in CD4^+^ T cells producing IL-17 only (Figure 4A), which was sustained at 24 h (Figure 4D). On the other hand, the CD4^+^ T cells producing IL-22 only showed an early trend towards higher response in the Trivalent-FP-immunized Tg groups (Figure 4B), with the response becoming statistically significant at 24 h (Figure 4E). Moreover, Trivalent-FP also induced IL-17 and IL-22 both producing CD4^+^ T cells (Figure 4D,F), which was statistically significant at 24 h (Figure 4F). For the most part, the induction of IL-17/IL-22 CD4^+^ T cells exhibited human-FcγRI dependency (Figure 4A–F). Interestingly, at 24 h post-infection, the Trivalent-FP induced IL-22-producing CD4^+^ T cells in the WT group as well, suggesting that the IL-22 production is not controlled by the human-FcγRI (Figure 4D). 

### 3.4. Receptor-Mediated Internalization and Antigen Presentation Was Enhanced by hFcγRI Targeting

Previous reports have demonstrated that crosslinking of the CD64 (FcγRI) leads to internalization of the receptor [23,24]. Accordingly, we anticipated that the anti-hFcγRI antibody fragment (scFv) in Trivalent-FP would interact with the hFcγRI on the APCs and facilitate antigen uptake and processing. To test this, splenocytes of Tg and WT mice were treated with Trivalent-FP at 4 °C and 37 °C. A correlative decrease in the hFcγRI mean fluorescence intensity (MFI) was observed in the Tg group, with increasing doses of Trivalent FP at 37 °C (Figure 5A). At 4 °C, however, there was no reduction in the levels of MFI of hFcγRI of APCs treated with Trivalent-FP (Figure 5A). Moreover, there was no change in the hFcγRI-MFI in analogous WT groups at any tested concentrations of FP (Figure 5A). This suggests that Trivalent-FP modulated the hFcγRI expression on the APC surface by receptor-mediated internalization of Trivalent-FP.

For evaluation of antigen presentation, we co-cultured macrophages with PspA-specific T cells, which produce IL-2 in response to PspA. We observed a dose- and hFcγRI-dependent increase in the IL-2 production associated with Tg APCs, in comparison with WT group (Figure 5B), suggesting that hFcγRI-mediated uptake enhances Ag processing and presentation. 

### 3.5. Trivalent-FP Induced Secretion of Pro-Inflammatory Cytokines by Lung Cells

Importantly, Trivalent-FP, which is an APC-targeted vaccine, did not require adjuvants to generate adequate immunity. Thus, we wanted to know if Trivalent-FPs themselves induce any adjuvant-like effects, which compensates the lack of adjuvants in this vaccine formulation. When adjuvants are injected, a local inflammatory response is usually induced, which is characterized by the presence of pro-inflammatory cytokines and chemokines and an influx of leukocytes. Furthermore, APCs are activated under the influence of these cytokines/chemokines and/or by direct activity of adjuvants. Activated APCs then take up Ag(s) and migrate to the draining lymph nodes where processed Ag(s) are presented to T cells. Thus, pro-inflammatory cytokines play an important role in orchestrating an immune response to vaccine Ag(s), including induction of leukocyte influx and APC maturation. 

To investigate whether Trivalent-FP can induce pro-inflammatory cytokines, lung cells from WT and Tg mice were harvested and treated with Trivalent-FP. Trivalent-FP induced significantly higher levels of IL-1α (Figure 6B), MIP-1α (Figure 6D), and TNF-α (Figure 6F) in the lung cells of the Tg groups. However, similarly treated WT lung cells exhibited no significant difference in cytokine production compared to the respective controls (Figure 6A,C,E), although these cells were perfectly capable of producing the respective cytokines in response to LPS (Figure 6A,C,E). This suggests that hFcγRI targeting of FP plays a role in cytokine induction by lung cells.

### 3.6. Trivalent-FP Induced Recruitment of Leukocytes and APC Activation in the Nasal-Associated Lymphoid Tissue 

In this investigation, the intranasal route was utilized for immunization, and in this context the NALT was important for immune activation. We anticipated that Trivalent-FP-induced pro-inflammatory cytokines may induce the recruitment of leukocytes in the NALT. To investigate this, mice were immunized with PBS or Trivalent-FP, and at 2 h after immunization mice were euthanized and their NALT was removed for cellular analysis. Trivalent-FP induced an increase in the total cell numbers in the NALT in both WT and Tg groups. However, the increase was more pronounced in the Tg group (Figure 7A). Upon further evaluation of cell types recruited, we found that DC (CD45^+^CD11c^+^) (Figure 7B), macrophages (CD45^+^CD11b^+^F4/80^+^) (Figure 7C), and neutrophils (CD45^+^Gr1^+^SSC^high^) (Figure 7D) were recruited in response to Trivalent-FP immunization, wherein more DCs were recruited in the Tg mice (Figure 7B) but more PMNs were recruited in WT mice (Figure 7D). In contrast, macrophage recruitment was comparable in both WT and Tg mice (Figure 7C). 

Following the analysis of leukocyte recruitment, we investigated the impact of Trivalent-FP on APC maturation in the NALT. At 24 h post-immunization, NALT cells were isolated and the level of MHC-II, and co-stimulatory molecule expression on the APCs were evaluated. We found a significant increase in the MFI of MHC-II (Figure 8A) and co-stimulatory molecules CD80/86 (Figure 8B) in the Tg group immunized with Trivalent-FP compared to that of the PBS-immunized group. The MFI for these markers were also significantly higher in the Tg group receiving Trivalent-FP as compared to the similarly immunized WT group (Figure 8A–F). This suggests hFcγRI involvement in the induction of increased MHC-II and co-stimulatory molecule (CD80/86) expression, as well as further supporting the notion of the induction of adjuvant-like effects by the hFcγRI-targeted FP. 

## 4. Discussion

In this study, we showed that Trivalent-FP induced protective immune response characterized by resistance against lethal challenge with *S. pneumoniae* (Figure 1) and systemic (Figure 2) and mucosal antibody response (Figure 3). The modification from Bivalent-FP to Trivalent-FP improved the vaccine efficacy and reduced the number of immunizations required to achieve >85% protective efficacy (Figure 1 and Figure 2) [13]. Moreover, it was also observed that induction of protective immunity by Trivalent-FP was associated with human-FcγRI (Figure 1 and Figure 2). 

Due to their central role in generating antigen-specific adaptive immune response, APCs have been targeted for vaccine development, utilizing various APC such as Toll-like-receptors (TLRs), CD11c, CD205, and MHC-II [25,26]. FcγR has also been targeted via the Fc domain of IgG in various studies. However, the Fc domain can be recognized by FcγRI, RIII, as well as RIIB. Although FcγRI and RIII engagement leads to DC activation, FcγRIIB generates an inhibitory signal [27]. The use of an antibody that specifically recognizes FcγRI avoids FcγRIIB engagement, thereby preferentially selecting for FcγR activation. Moreover, FcγRI is exclusively expressed on macrophages and DCs, which makes it suitable for targeting specific APCs [28]. In this investigation (Figure 1, Figure 2 and Figure 3) we also observed that antigen-specific immunity can be generated by targeting the antigen to APCs via FcγRI, which is consistent with our previous report [13]. Furthermore, Keler et al. previously demonstrated that a trivalent version of a human FcγRI-specific antibody induced better immune response in vivo than its bivalent counterpart [29]. Consistently, in this study we observed improvement in the generation of pathogen-specific protective immune responses by increasing the valency of the hFcγRI-specific single-chain antibody from dimeric to trimeric (Figure 1, Figure 2 and Figure 3).

Secretory IgA plays a critical role in protection against various viral and bacterial infections at the mucosal surfaces [30,31,32,33,34,35]. For example, dimeric IgAs block interaction of pathogens with epithelial cells and prevent trans-epithelial invasion of pathogens [36,37,38,39,40,41,42,43]. Furthermore, the critical role of secretory IgA has been demonstrated in several studies [44,45,46,47]. In this investigation, we demonstrated that *S. pneumoniae*-specific IgA is induced by Trivalent-FP (Figure 3A,B), which is consistent with our previous report using Bivalent-FP [13]. Multiple studies have also shown that targeting antigens to FcγRI induces enhanced Ag-specific Ab responses [13,20,29], which corroborates our observations in this study. 

The critical role of IL-17 in mucosal protection against *S. pneumoniae* has been well documented in many studies [47,48,49,50,51,52]. Specifically, IL-17 has been implicated in cross-protection against several *S. pneumoniae* serotypes [53,54,55,56]. In this investigation, Trivalent-FP induced IL-17, producing CD4^+^ T cells early on (6 h) which was sustained at 24 h (Figure 4A,D). Moreover, Trivalent-FP also induced IL-22-producing CD4^+^ T cells (Figure 4B,E). Although IL-17 response by Trivalent-FP exhibited human-FcγRI dependency, the IL-22 response seemed independent of the receptor (Figure 4E). However, it remains to be investigated if the IL-17 and IL22 response by Trivalent-FP is regulated by human-FcγRI. Interestingly, IL-17 and IL-22 are known to play important roles in protection against several pathogens that infect via mucosal routes [19,57,58,59], which suggests that the Trivalent-FP platform may be useful in developing vaccines against many other mucosal pathogens.

Subunit vaccines that are approved for human use to date require adjuvants to enhance their immunogenicity. Notably, despite the lack of a conventional adjuvant in our anti-hFcγRI-based vaccines, we observed a robust mucosal and systemic immune response and enhanced protection against the cognate pathogen [13] (Figure 1, Figure 2 and Figure 3). This then raised the important question as to whether Trivalent-FP itself induces adjuvant-like activity. Adjuvant activity, although poorly defined, includes recruitment of neutrophils, DCs, and macrophages, as well as production of pro-inflammatory cytokines. More importantly, adjuvant activity modulates APC functions by increasing the expression of co-stimulatory molecules. In this regard, we observed the fact that Trivalent-FP can induce pro-inflammatory cytokines in mucosal tissues (Figure 6). Moreover, we also observed an influx of PMN, macrophages, and DCs in the NALT following intranasal immunization (Figure 7). Importantly, leukocyte recruitment is likely affected by the cytokines released by the action of Trivalent-FP, because IL-1α, which is induced by Trivalent-FP (Figure 6B), plays a critical role in recruitment of leukocytes [60,61]. Adjuvant activity modulates APC functions by increasing the expression of co-stimulatory molecules. In this study, we observed that intranasal immunization with Trivalent-FP induced up-regulation of APC maturation markers, including MHC-II (Figure 8A) and CD80/86 (Figure 8B). In addition, APC activation can occur in presence of TNF-α [62], which is released following administration of Trivalent-FP (Figure 6F). It is important to note that other studies have also shown that Ab and cell-mediated immune responses can be generated by MHC-class II, CD-40, and CD207 DC targeted vaccines, even in the absence of adjuvants [63,64]. It has also been shown that engagement of FcγRI and III triggers DC activation, inducing the production of various cytokines and chemokines, as well as changes in expression of cell surface proteins involved in Ag presentation [65,66,67]. However, additional studies will be needed to fully understand the mechanistic aspects of FP-induced adjuvant activity. 

We also observed enhanced uptake of FPs (Ag) by hFcγRI-expressing APC, which correlated with an enhanced Ag processing and presentation (Figure 5A,B). This can be explained in part by the fact that Ags are taken up and presented more efficiently by hFcγRI-expressing APCs compared to WT APCs. Many studies have shown that FcγRI crosslinking results in enhanced uptake by receptor-mediated internalization [68,69,70] and enhanced Ag presentation [71,72,73].

It is also important to note that stimulation of the mucosal immune response is preferable to protect against pathogens that enter via oral and pulmonary routes. Specifically, direct exposure of the mucosal immune system to immunogens induces more potent mucosal immunity compared to parenteral vaccination. Although direct mucosal immunization with live attenuated vaccines has been successful, the use of subunit vaccines via this route has been proven to be problematic. The dilution of the vaccine Ags in the mucosal fluids is one of the current challenges, whereas mucous fluid may also prevent contact between the vaccine components and the mucosal epithelium [74]. In this regard, induction of protective immune response by mucosal immunization using our hFcγRI-targeted vaccine platform is noteworthy, more so because it is based on a subunit vaccine platform and does not require the addition of traditional adjuvants. Using a similar approach Ye and coworkers [75] have shown that intranasal immunization can induce an effective immune response against Herpes Simplex Virus (HSV). Specifically, a subunit vaccine consisting of the HSV-2 envelope glycoprotein fused to the IgG Fc fragment was delivered intranasally, eliciting systemic, as well as mucosal, B and T cell responses and conferred protection from intravaginal challenge with HSV-2. However, in this study, investigators used CpG as mucosal adjuvant and, unlike our FP vaccine, this vaccine lacked FcγR specificity, in that it could also engage multiple FcγR types including the inhibitory FcγRIIB. 

## 5. Conclusions

In conclusion, the intranasal immunization with Trivalent-FP induced recruitment of leukocytes, including APCs such as DCs and macrophages, which took up, processed, and presented the cognate antigen (PspA). Antigen processing and presentation was facilitated by FP engagement of hFcγRI. As a result, T cell and B cell responses ensued, characterized by enhanced humoral (IgG and IgA) and cell-mediated immunity (Th17 and Th22). Consequently, hFcγRI-expressing Tg mice receiving Trivalent-FP exhibited greater survival and resistance against *S. pneumoniae* infection. The fact that this vaccine platform induced a robust humoral and cell-mediated immune response at a mucosal site makes it an attractive platform for mucosal vaccine development against other mucosal pathogens. Specifically, this platform can also be utilized for vaccines against pathogens such as influenza (antigens: Hemagglutinin, Neuraminidase) [76,77], *Clostridium difficile* (sntigens: TcdA, TcdB) [78], and enterotoxigenic *Escherichia coli* (antigens: colonization factor antigens, heat-stable toxin, heat-labile toxin) [79], which have well-defined protein antigens that can be readily fused with the APC-targeting component of Trivalent-FP. Importantly, most of these antigens (TcdA, TcdB, heat-stable toxin, and heat-labile toxin) are toxins, and are genetically modified to minimize their toxicity in order to utilize them in a vaccine. However, in doing so, these antigens lose their immunogenicity. We anticipate that by fusion of these antigens with the APC-targeting component of Trivalent-FP, their immunogenicity can be significantly increased, and they will not require additional adjuvant for induction of adequate immunity. 

## Figures and Tables

**Figure 1 vaccines-08-00193-f001:**
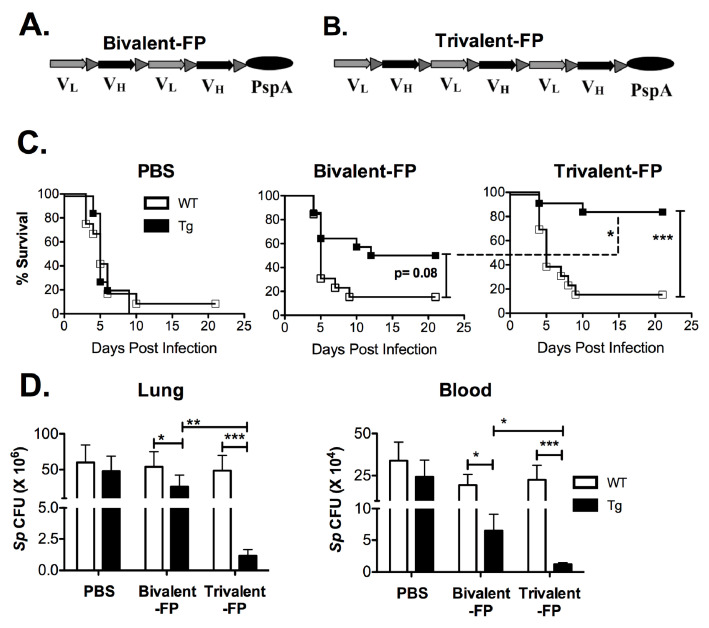
Trivalent-FP induced enhanced protection against pneumococcal infection compared to Bivalent-FP: (**A**,**B**) Schematic representation of the vaccines used in this study. Both vaccines contained a single chain variable fraction antibody (ScFv) and the antigen pneumococcal surface protein A (PspA). VL and VH represent the light and heavy chains of the ScFv, respectively. Bivalent-FP (A) contains two pairs of the ScFv, whereas Trivalent-FP (B) contains three pairs, and both have one copy of the antigen PspA. (**C**,**D**) Groups of WT (wild type) and Tg (transgenic) mice were immunized twice at an interval of 3 weeks with PBS, Bivalent-FP (208 pmol), or Trivalent-FP (208 pmol) via the intranasal route, and challenged with a lethal dose (2 × 10^6^ CFUs) of *Streptococcus pneumoniae* at 2 weeks post-booster immunization. (**A**) Kaplan–Meier survival curve is presented; combined data from two independent experiments is shown (*n* = 14/group, *** *p* = 0.005). Statistical significance between indicated groups was evaluated by Mentel–Cox (log-rank) test. (**B**) Following immunization and challenge, bacterial burden (*Sp* colony forming unit (CFU): *S. pneumoniae* CFU) in blood and lung homogenates was evaluated on day 4 post-infection. Mean ± SE of data from two independent experiments is shown (*n* = 10/group, * *p* = 0.05, ** *p* = 0.01, *** *p* = 0.005). Statistical significance between indicated groups was evaluated by Mann–Whitney nonparametric test.

**Figure 2 vaccines-08-00193-f002:**
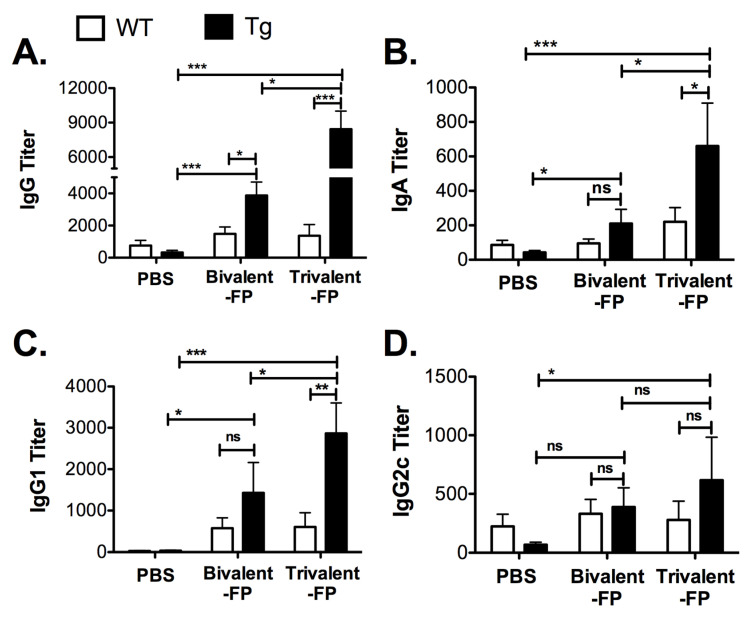
Trivalent-FP induced an *S. pneumoniae*-specific humoral immune response: groups of WT (wild type) and Tg (transgenic) mice were immunized with PBS, Bivalent-FP (208 pmol), or Trivalent-FP (208 pmol) via the intranasal route, and serum was obtained 11 days post-booster immunization. *S. pneumoniae-*specific IgG (**A**), IgA (**B**), IgG1 (**C**), and IgG2c (**D**) titers were evaluated by ELISA. Mean ± SE of data from two independent experiments is shown (*n* = 12/group, * *p* = 0.05, ** *p* = 0.01, *** *p* = 0.005, ns: not significant). Statistical significance between indicated groups was evaluated by Mann–Whitney nonparametric test.

**Figure 3 vaccines-08-00193-f003:**
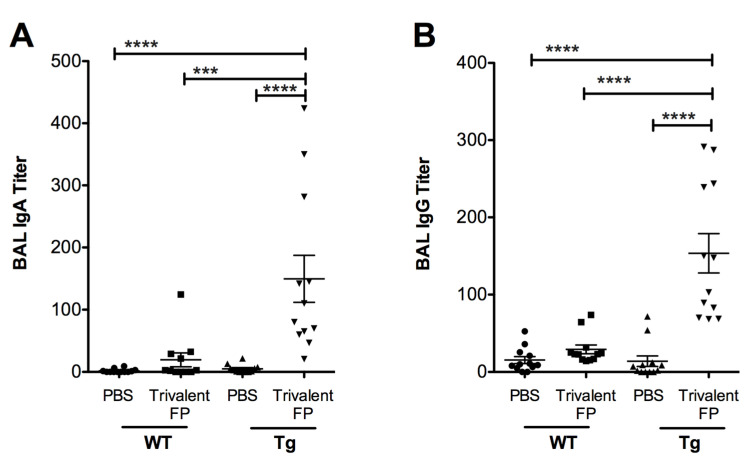
Trivalent-FP induced a mucosal antibody response: groups of WT (wild type) and Tg (transgenic) mice were immunized with PBS or Trivalent-FP (208 pmol) via the intranasal route, and bronchoalveolar lavage (BAL) was collected at 14 days post-booster immunization. *S. pneumoniae-*specific IgA (**A**) and IgG (**B**) titers in BAL were evaluated by ELISA. Mean ± SE of data from two independent experiments is shown (*n* = 12/group, *** *p* = 0.001, **** *p* = 0.0001). Statistical significance between indicated groups was evaluated by Mann–Whitney nonparametric test.

**Figure 4 vaccines-08-00193-f004:**
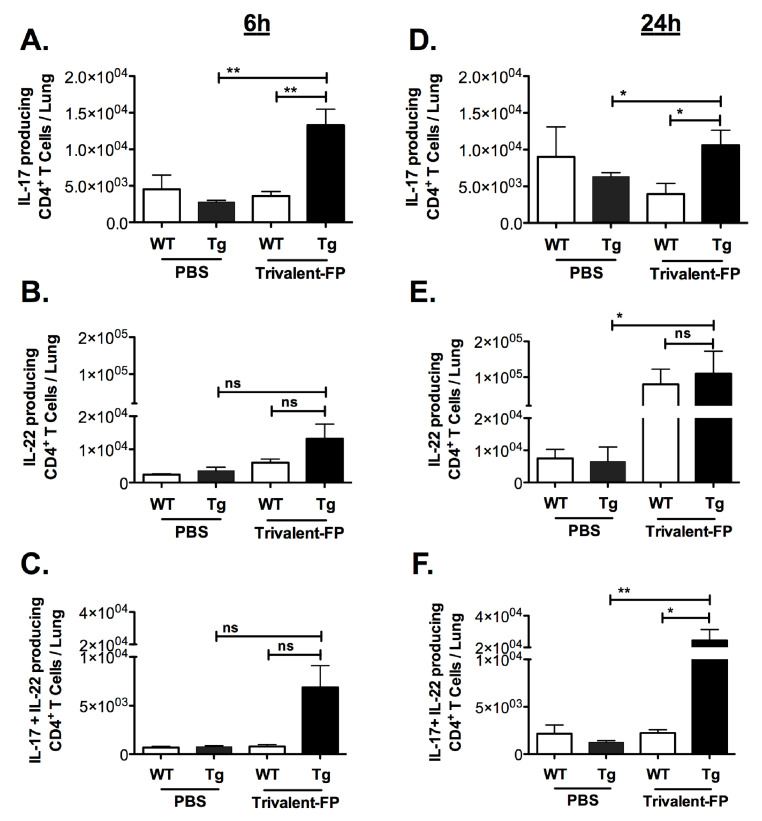
Intranasal immunization with hFcγRI-targeted PspA induced pulmonary antibody and cell-mediated immunity: groups of WT (wild type) and Tg (transgenic) mice were immunized with PBS or Trivalent-FP (208 pmol) via the intranasal route and infected with *S. pneumoniae* 6 weeks post-booster immunization. At 6 h (**A**–**C**) and 24 h (**D**–**F**) post-infection, mice were euthanized and their lungs were excised. Single cells were isolated from lungs and stained with CD45-antigen-presenting cell (APC)-Fire-750, CD3-V450, CD4-FITC, IL-17-PE, and IL-22-APC. By flow cytometry analysis, single cells were discriminated and further gated on the CD3^+^ CD4^+^ population, and the CD3^+^CD4^+^ T cells producing IL-17 alone (**A**,**B**), IL-22 alone (**C**,**D**), or both IL-17 and IL-22 (**E**,**F**) were enumerated. Mean ± SE of data from two independent experiments is shown (*n* = 6/group, * *p* = 0.05, ** *p* = 0.01, ns = not significant). Statistical significance between indicated groups was evaluated by Mann–Whitney nonparametric test.

**Figure 5 vaccines-08-00193-f005:**
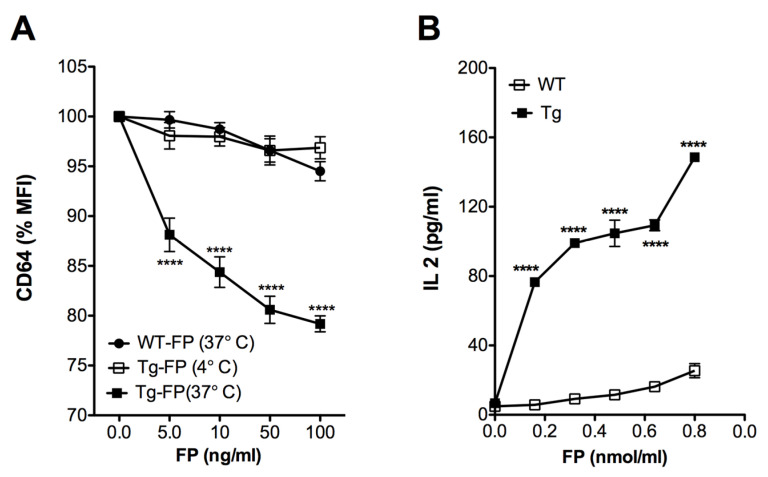
Human FcγRI targeting facilitated antigen uptake and presentation: (**A**) splenocytes from WT (wild type) and Tg (transgenic) mice were treated with the indicated amounts of Trivalent-FP at 4 °C or 37 °C for 2 h and stained with CD19-PE, CD11b-APC, CD11c-BV421, and anti-human and anti-mouse CD64-FITC antibodies. CD64 expression on the surface of CD19^−^ CD11b^+^ CD11c^+^ cells was evaluated by flow cytometry. Percentage change in the MFI relative to untreated cells is shown. (**B**) PspA-specific T cell hybridoma B6D2 was co-cultured with WT or Tg peritoneal exudate macrophages and treated with indicated amounts of Trivalent-FP. Then, 48 h following incubation, IL-2 levels were measured in culture supernatants by biplex assay. Mean ± SE of data from two independent experiments is shown (*n* = 6/group, **** *p* = 0.0001). Statistical significance between groups was evaluated by Mann–Whitney nonparametric test.

**Figure 6 vaccines-08-00193-f006:**
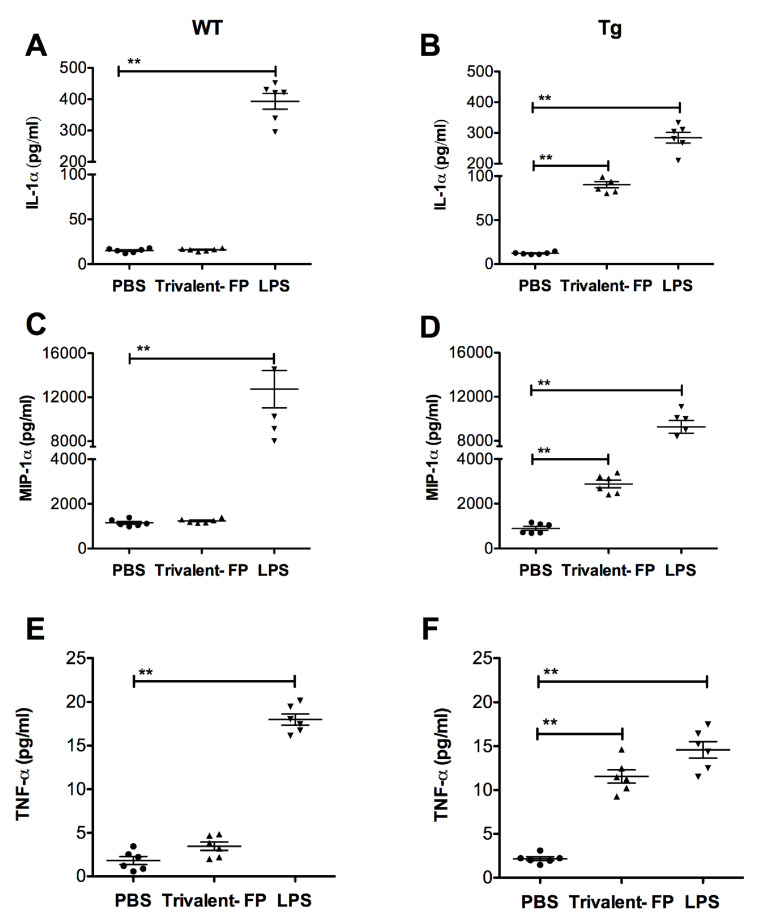
Trivalent-FP induced cytokine release by pulmonary cells in a hFcγRI-dependent manner: single cell preparation from lungs of WT (wild type) (**A**,**C**,**E**) or Tg (transgenic) (**B**,**D**,**F**) mice were treated with PBS, 50 μg Trivalent-FP, or 100 ng lipopolysaccharide. Forty-eight hours following the treatment, cytokines were evaluated in culture supernatants using bioplex multiplex assay. Mean ± SE of data from two independent experiments is shown (*n* = 6/group, ** *p* = 0.01). Statistical significance between indicated groups was evaluated by Mann–Whitney nonparametric test.

**Figure 7 vaccines-08-00193-f007:**
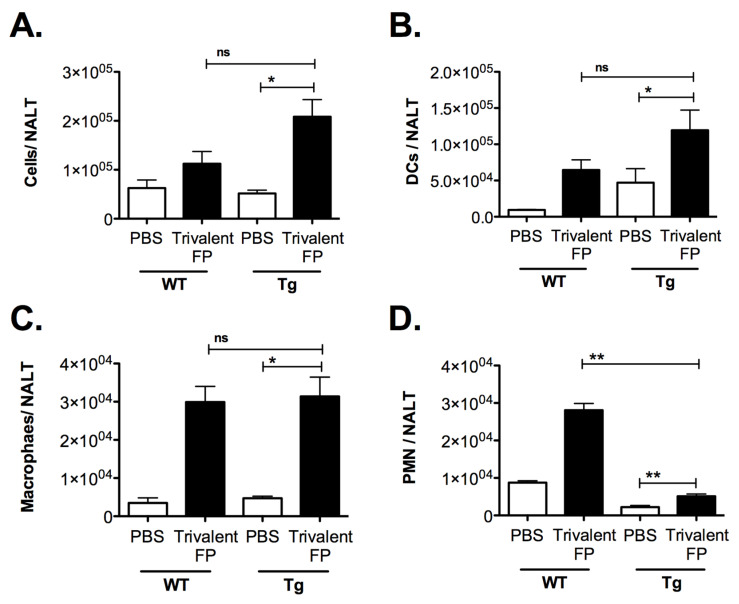
Trivalent-FP induced recruitment of leukocytes into the nasal-associated lymphoid tissue (NALT): groups of WT and Tg mice were immunized intranasally with PBS or Trivalent-FP. Two hours post-immunization, NALT cells were isolated and total cells were enumerated by microscopy (**A**). Subsequently, cells were stained with CD45-APC-Fire-750, CD11b-FITC, CD11c-APC, Gr1-Pacific blue, and F4/80- PE. CD45-positive cells were further evaluated, and DC (CD11c^+^) (**B**), macrophages (CD11b^+^, F4/80^+^), (**C**) and PMN (Gr1^+^, SSC^high^), (**D**) were enumerated. Various cell populations were normalized to single cells, excluding doublets. Mean ± SE of data from two independent experiments is shown (*n* = 6/group, * *p* = 0.05, ** *p* = 0.01, ns = not significant). Statistical significance between indicated groups has been evaluated by Mann–Whitney nonparametric test.

**Figure 8 vaccines-08-00193-f008:**
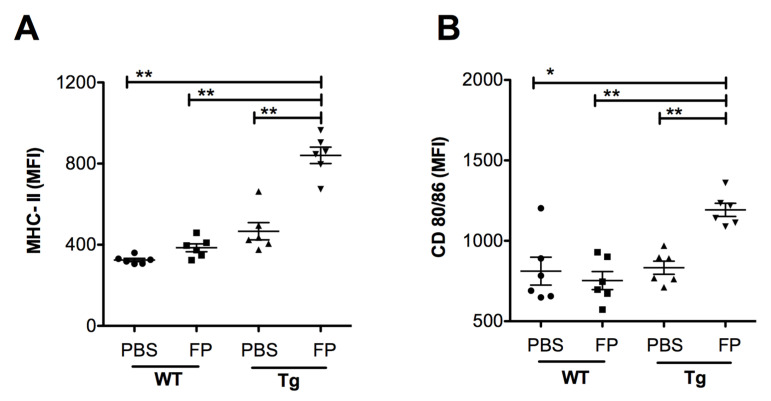
Trivalent-FP induced APC activation in the NALT of Tg mice: groups of WT (wild type) and Tg (transgenic) mice were immunized intranasally with 20 μL PBS or 20 μg Trivalent-FP (in 20 μL PBS). Twenty-four hours after immunization, mice were euthanized and their NALT cells were isolated. Cells were stained with CD11b-PE-Cy7, CD11c-APC, MHC-II-FITC, CD19-PE, CD80-PerCP-Cy-5.5, CD86-PerCP-Cy-5.5, and Fixable Viability Dye eFluor 780. Viable cells were gated on CD19^−^ and MHC-II^+^ and the mean fluorescence intensity (MFI) of MHC-II (**A**) and CD80/86 (**B**) was calculated. Mean ± SE of data from two independent experiments is shown (*n* = 6/group, * *p* = 0.001, ** *p* = 0.01). Statistical significance between indicated groups was evaluated by Mann–Whitney nonparametric test.

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
