# Peer review of "Preclinical Efficacy of a Trivalent Human FcγRI-Targeted Adjuvant-Free Subunit Mucosal Vaccine against Pulmonary Pneumococcal Infection"

_vaccines, 2020, doi:10.3390/vaccines8020193_

Round 1

Reviewer 1 Report

The paper by Kumar, S. is interesting and will really bring deep knowledge in the field if represented properly. I only have a problem with their choice of microbe (S. pneumoniae). There is no explanation given why this particular microbe being used?

Also, can this system be expanded to other microbes? If it is not possible to explain this experimentally, the authors can at least hypothesize and this will also increase the readership of the journal.

Author Response

Response to Reviewer 1 Comments

I only have a problem with their choice of microbe (S. pneumoniae). There is no explanation given why this particular microbe being used?

Response: With over 90 known serotypes, S pneumoniae remains the leading cause of community acquired pneumoniae. The younger and immunocompromised populations are more vulnerable to get this infection and it causes over a million deaths per year in children aged less than 5 years. To date, two polysaccharide-based subunit vaccines are available to combat select serotypes. However, use of these 13 and 23 serotype vaccines cause serotype replacement in the vaccinated population. This results in a surge of non-vaccine serotypes over the vaccinated population. Thus, new approaches to pneumococcal vaccines are required, which can generate protection over multiple serotypes. In addition, the pneumococcal surface protein A (PspA) has been known to induce cross-serotype protection against S pneumoniae. Thus, S pneumoniae is not only important for global health, it also possesses a well-defined and relatively conserved surface protein antigen (PspA), making it an ideal model to test our human FcgRI-targeted mucosal subunit vaccine. We have added the above text in the introduction (Lines 48-58) to clarify the choice of S pneumoniae as a model in this study.

Also, can this system be expanded to other microbes? If it is not possible to explain this experimentally, the authors can at least hypothesize, and this will also increase the readership of the journal

Response: We have added text (lines 530-541) pointing out the potential use of this platform for vaccines against other mucosal pathogens such as enterotoxigenic E coli, Clostridium difficile, and influenza.

Reviewer 2 Report

Major:

  1. The study design looks confusing. Author must add positive control for all the figures as they did for Figure 1&2. Without adding Bivalent-FP, the efficacy of Trivalent can't be determined.
  2. Authors should clearly mention the significance of Th17 cell type in the introduction section. Why not Effector or memory CD4+ T-cells? why CD4+ only? Why not CD8+ T-cells as well.
  3. Authors should provide rational for acquiring the cells at various time points in the figures. It makes very confusing the way they explained.
  4. Line no. 148-152: It is advised to clearly explain how this markers differentiate cell types. (eg. Monocyte/macrophage: X-Y+Z-).
  5. It is advised to add how Trivalent-FP was generated? plasmid? If so then which site of plasmid. As a first time reader nobody will go back to their 2012 paper to see how this was prepared. The methodology section should speak up atleast brief explanation if it is continuation of previous study.
  6. The significance level annotation should be consistent throughout the figures. Comparison of different groups should be provided with different sign of annotations.
  7. Authors have deliberately increases the figures. Figure 4. A, B, and C can be combine together and D,E,F together. Please consider this suggestion for Figure 6 as well.
  8. Figure 7. Are these cell population counts normalized to starting point? If not then it should be against CD45+ cells/or single cells after removing doublets. Or authors can compare the cell counts against CD45+ positive cells (if they did it magnetic bound antibodies) if they did it before flow acquire.
  9. It is advised to develop all the figure legends. Legends and Figures should make understand the findings of the study. Authors should explain abbreviations and statistics as well.

Minor

  1. Author must spell out the full form of abbreviations in on their first appearance (eg. NALT).
  2. Typographical errors throughout the manuscript.

Author Response

Response to Reviewer 2 Comments

Major

The study design looks confusing. Author must add positive control for all the figures as they did for Figure 1&2. Without adding Bivalent-FP, the efficacy of Trivalent can't be determined.

Response: In this investigation we first evaluated the protective efficacy of the Trivalent-FP by comparing it with the Bivalent-FP. From the observations in figures 1 and 2, it was clear that the Trivalent-FP is more efficacious than the Bivalent-FP. Then, in the later part of this investigation we wanted to evaluate the mucosal cell-mediated immune responses and the innate immune responses pertaining to adjuvant effects induced by targeting PspA to human FcgRI mucosally. In this case, we reasoned that since the Trivalent-FP is more efficacious, it would yield a greater overall signal relevant to cell-mediated and innate immune responses generated by Bivalent FP. Therefore, we focused our study on the Trivalent-FP at this point. We have added additional text in the Introduction (72-86) and at the beginning of Result-3 (297-299), which more clearly explains our choice of this experimental design.

Authors should clearly mention the significance of Th17 cell type in the introduction section. Why not Effector or memory CD4+ T-cells? why CD4+ only? Why not CD8+ T-cells as well.

Response: We agree on this point. Effector or memory CD4+ T and CD8+ cells do have a central role in vaccine-induced long term protection. However, emerging evidence suggests that Th17-polarized CD4+ T cells play important roles in mucosal immunity against many pathogens. Moreover, Th17 is also pertinent to S pneumoniae, the model pathogen used in this study. We have thus added text clarifying this in the Introduction (lines 77-83). Importantly, in this regard, future studies involving multiple pathogens will further focus on the other memory and effector CD+/CD8+ cells.

Authors should provide rational for acquiring the cells at various time points in the figures. It makes very confusing the way they explained.

Response: S pneumoniae causes an acute infection wherein mice become moribund within 3-4 days, so it is anticipated that the protective response must be elicited very early. On the other hand, it is also known that S pneumoniae infection by itself causes recruitment of Th17 and Th22 cells with a delayed kinetics. Thus, we anticipated that the immunized mice would recruit Th17 and Th22 cells with faster kinetics. Therefore, we chose an immediate early time point (6h) and an early time point (24h). We have added our explanation of the choice of these time points to the text (Lines 317-324).  

Line no. 148-152: It is advised to clearly explain how this marker differentiate cell types. (eg. Monocyte/macrophage: X-Y+Z-).

Response: Lines 191-194 have been added to explain the monocyte/macrophage discrimination by the selected markers. 

It is advised to add how Trivalent-FP was generated? plasmid? If so, then which site of plasmid. As a first-time reader nobody will go back to their 2012 paper to see how this was prepared. The methodology section should speak up at least brief explanation if it is continuation of previous study.

Response: The generation of Trivalent-FP is now described in more detail (Section 2.3 Vaccine Preparation, Lines 105-114). 

The significance level annotation should be consistent throughout the figures. Comparison of different groups should be provided with different sign of annotations.

Response: The significance level annotation has now been made consistent throughout the figures.

Authors have deliberately increases the figures. Figure 4. A, B, and C can be combine together and D,E,F together. Please consider this suggestion for Figure 6 as well.

Response: This is correct. Thus, as suggested, we did try combining these figures as suggested. However, the scales differ to such a degree that some bars become barely visible when figures are combined. We felt figures in their current form allow readers to more easily distinguish and interpret differences in the data.

Figure 7. Are these cell population counts normalized to starting point? If not then it should be against CD45+ cells/or single cells after removing doublets. Or authors can compare the cell counts against CD45+ positive cells (if they did it magnetic bound antibodies) if they did it before flow acquire.

Response: Yes, the cell populations are normalized to single cells, excluding doublets. A clarification in this regard has been added in the figure legends section (Lines 418-419).

It is advised to develop all the figure legends. Legends and Figures should make understand the findings of the study. Authors should explain abbreviations and statistics as well.

Response: As recommended, we have further elaborated in the figure legends to further facilitate understanding of the findings. Statistical methods have also been further described and the abbreviations have been explained.

Minor

Author must spell out the full form of abbreviations in on their first appearance (eg. NALT).

Response: This has been corrected.

Typographical errors throughout the manuscript.

Response: This has been corrected.

Round 2

Reviewer 2 Report

I have gone through all the modifications in the current version of manuscript. Authors responded all the queries and improved significantly. I recommend to accept the manuscript.